# Impact of the Combination of Probiotics and Digital Poultry System on Behavior, Welfare Parameters, and Growth Performance in Broiler Chicken

**DOI:** 10.3390/microorganisms11092345

**Published:** 2023-09-19

**Authors:** Victor A Zammit, Sang-O Park

**Affiliations:** 1Metabolic Biochemistry, Warwick Medical School, University of Warwick, Coventry CV4 7AL, UK; vic.zamit01@gmail.com; 2HooinEcobio Institute, Hongseong-gun 32280, Chungcheongnam-do, Republic of Korea

**Keywords:** broiler chicken, behavior welfare, digital poultry system, growth performance, probiotics

## Abstract

Recently, applied technology in the form of the combination of a probiotics and a digital poultry system, with the convergence of Information and Communications Technology and farm animals, has enabled a new strategy to overcome the livestock production crisis caused by climate change, while maintaining sustainable poultry farming in terms of care, feeding, and environmental management systems for poultry. The aim of this study was to investigate the biological mechanisms of animal behavioral welfare and production improvement using the combination of a probiotics and a digital poultry system in broiler chickens. A total of 400 one-day-old male broilers (ROSS 308) were randomly divided into four treatment groups, with five replicates each (20 birds/replicate pen) in a completely randomized design: control group with a conventional poultry system without probiotics (CON), conventional poultry system with 500 ppm of probiotics (CON500), digital poultry control system without probiotics (DPCS), and digital poultry system with 500 ppm of probiotics (DPS500). All experimental animals were reared for 35 days under the same standard environmental conditions. The experimental results indicated that the animal behavioral welfare, which includes drinking, eating, locomotion, grooming, and resting, in addition to foot pads, knee burns, plumage, and gait scores, as well as the growth performance of the broiler chickens, were improved by maintaining immune function and cecal microbiota balance via interaction between the combination of a probiotics and a digital poultry system. In conclusion, it was found that the combined system showed improved broiler growth performance and animal behavioral welfare. Thus, further studies of molecular biological mechanisms by the use of such a combined system to improve the nutritional composition and quality of chicken meats are recommended.

## 1. Introduction

The changes to animal feeding management and environmental conditions arising from global warming can increase animal stress levels greatly and affect animal behavioral welfare, which can lead to the loss of farm animals and food crises [1]. Probiotics are widely known to improve behavioral welfare and productivity in farm animals, while simultaneously lowering animal stress under poultry feeding management and environmental conditions [2,3]. A probiotic composed of *B. subtillus*, *S. galilaeus*, and *Sphingobacteriaceae* has been reported to improve egg production, egg quality, and broiler production, while reducing odor [4,5].

Probiotics and digital livestock systems have been introduced as sustainable livestock strategies to help combat climate change [1,6]. Such technologies can improve livestock production, as well as farm income through enhanced animal health, behavior, and welfare while reducing environmental issues due to climate change [7,8].

In some previous studies, digital livestock systems were reported for improving egg production, egg quality, and animal behavioral welfare through increased nutrient digestibility, as well as balanced blood parameters, immune function, cecal microbiota, and short-chain fatty acids in laying hens and swine [6,9]. The digital poultry system is an innovative animal-feeding and -management model that can meet the constraints of climate change by using remote control, sensing, and precision livestock technology for animal movements and management and environmental conditions such as temperature, humidity, ventilation, and smell [10,11]. It is expected to reduce the time and labor necessary for feeding management involving water and feed intake, weight, litter, and dead chicks, while boosting animal health and facilitating business management [12,13]. Owing to these benefits, researchers in many countries have studied diverse application methods to increase the production, behaviors, and welfare of broiler chickens using digital poultry systems [1,10,14].

The health of broiler chickens depends primarily on animal management and environmental condition control. As factory-intensive conventional livestock systems have fallen behind in terms of automation and insufficient labor, the overall control of poultry houses has been unsatisfactory. Moreover, a poor environment has negative effects on the animal behavioral welfare, which results in increased animal stress and impaired welfare, leading to reduced poultry production [15,16,17].

Until recently, little has been reported in the literature about the biological mechanism of the combination of a probiotics and a digital poultry system for improving animal behavioral welfare such as drinking, eating, locomotion, grooming, resting, and also foot pads, hock burns, plumage, and gait scores, in addition to growth performance such as body weight, feed intake, and the feed conversion ratio of broiler chickens [6,9,10,18].

However, based on the previously studied biological mechanism for improving the productivity and behavioral welfare of farm animals, applying the combination of a probiotics and a digital poultry system may have a positive effect on the growth performance and behavioral welfare of broiler chickens. Therefore, the study hypothesized that the application of this combined system will improve behavioral welfare and growth performance in broiler chickens. The aim of this study was to identify the biological mechanisms for improving immune functions, cecal microbiota balance, animal behavioral welfare, and growth performance in broiler chickens by using the combination of a probiotics and a digital poultry system.

## 2. Materials and Methods

### 2.1. Ethical Approval

This study was conducted according to the guidelines for scientific and ethical procedures provided in the European Experimental Animal Handling License (SCT-w94058) and European Union Council Directive 2008/120/EC. The Scientific Research Ethics Committee of the Ministry of SMEs and Startups (MSS) of the Republic of Korea approved all the animal tests conducted in this study (Ethics Reference No: E-2035).

### 2.2. Animals, Experimental Design, and Feeding Management

The animal experiments were performed in poultry research farms, the HooinEcobio Institute, Hongseong, Republic of Korea, 2021. Four-hundred healthy one-day-old male broiler chicks (Ross-308, BW 40.05 g) were purchased from Hanyang hatchery (Anseong-si, Republic of Korea). The animals were allocated between four treatment groups with five replicates each in a completely randomized design for 35 days as follows: control group with a conventional poultry system without probiotics (CON), conventional poultry system with 500 ppm of probiotics (CON500), digital poultry control system without probiotics (DPCS), and digital poultry system with 500 ppm of probiotics (DPS500). The animals were placed in five replicate pens with 20 broiler chickens per pen (stocking density of 10 birds/m^2^). The probiotic was prepared by combining 2% of *Bacillus subtillus*, *Streptomyces galilaeus*, and *Sphingobacteriaceae* (isolated from earthworm cast) with 98% earthworm cast. These products were provided by HooinEcobio Co., Ltd. (Hongseong, Republic of Korea, Korean Patent No. 0092670). Each strain contained at least 3.5 × 10^8^ colony-forming units per gram (CFU/g product) of the product [5]. The amount of probiotics added was determined based on previous results that demonstrated plateau levels without a further increase in growth performance under standard environmental conditions for broiler chickens in research farms. The experimental diets were formulated based on the nutritional requirements for broiler chickens as recommended by the NRC [19]. The formula and chemical compositions of the basal diets are shown in Table 1. The nutrient compositions of the experimental diets were analyzed in accordance with the AOAC testing methods [20]. The animals were fed a grower diet from Days 1 to 21 and a finisher diet from Days 22 to 35.

The optimal environmental conditions, including temperature, humidity, and lighting, were maintained according to the Ross strain guidelines, as well as the Korean standard brooding practices and standard operating procedures [4,21]. The CON and CON500 groups were reared separately with repeat pens for each treatment group under the same conventional livestock system. The DPCS and DPS500 groups were reared as described above, but using a digital livestock system. The two systems were housed in separate rooms in the same building. During the entire experimental period, the animals for all treatment groups were maintained under the same standard environmental conditions. Animals for the grower (1–21 days) and finisher (22–35 days) periods were reared under the same standard environmental conditions: temperature, humidity, and lighting (30–40 lux)/darkness maintained at 30–35 °C, 60–70%, and 24 to 22/2 h from the first day to one week; 30–25 °C, 50–60%, lighting (20–25 lux)/darkness 18/6 h for the grower period; and 22 °C, 50–55%, lighting/darkness 23/1 h for the finisher period, respectively. Using an installed ventilation controller, the minimum and maximum ventilations were adjusted to 20–30% and 60–70%, respectively, to minimize changes to the internal environment in line with changes to the external environment. The animals were reared in floor pens covered with a 5 cm-thick layer of wood shavings and allowed *ad libitum* access to diet and water. All birds were vaccinated against Newcastle disease (ND at 1, 14, and 22 days of age), Gumboro disease (infectious bursal disease virus (IBDV) at 21 days of age), and infectious bronchitis disease (IBD at 14 and 22 days of age) according to manufacturer instructions (Fort Dodge Animal Health, Overland Park, KS, USA).

### 2.3. Digital Poultry System

The study used a digital poultry system (Figure 1), which converges Information and Communications Technology (ICT) and farm animals through the use of monitors and remote control to check the movement and health of all animals in livestock herds, which included big data, monitoring sensors, an imaging system (CCTV), the Internet of Things (IoT), machine learning (ML), cloud computing, a personal computer, and mobile phones. It was equipped with various features such as automatic feeder and drinking water dispensers, temperature and humidity sensors, an odor removal system, cooling pads, and an air conditioner. The system was applied at the room level (odor control, air conditioning, etc.) in the same building. Through these systems, more data related to behavioral welfare and growth performance can be collected from each animal. For example, such data can be collected through intelligent CCTV (PRO-1080MSFB, Swann Communications, Santa Fe Springs, CA, USA), image recognition software (MATLAB-R2019b), wearable sensing, and body-weight- or sound-monitoring devices. The data can also help improve animal health through the monitoring of climate, air quality, ventilation, and animal movements.

### 2.4. Growth Performance and Immune Organ Index

Live body weights and feed intake were recorded weekly for 35 days of the experimental period. The growth performance was then calculated based on the body weight, feed intake, and feed conversion ratio (weight gain/feed intake) at the beginning and the end of the experiment (35 days). At the end of the feeding experiments, the chickens were euthanized using CO_2_ gas (concentration of 40% or less), and their immune organs and cecum were collected. The spleen, thymus, and bursa of Fabricius index (as g per 100 g live body weight) are indicated. The cecum was collected and frozen in liquid nitrogen for storage at −80 °C to perform the bacterial counts.

### 2.5. Immune Biomarkers

To evaluate the immune biomarkers of the animals, about 8 mL of blood was collected from the jugular vein of each bird at the end of the experiment and placed in a clot activator vacuum tube (Becton Dickinson Vacutainer Systems, Franklin Lakes, NJ, USA). The serum corticosterone (a stress hormone) level was quantified using a chicken corticosterone ELISA Kit (Mybioscience, Inc., San Diego, CA, USA) according to the manufacturer’s manual. The serum immunoglobulin G (IgG) level was determined using a chicken cortisol ELISA Kit (Mybioscience, Inc., San Diego, CA, USA). Three blood smears per animal were obtained immediately after the blood collection, and the slides were stained using May-Grunwald-Giemsa after air drying. The heterophils (H) and lymphocytes (L) were differentiated using light microscopy. After 100 leukocytes per slide and 200 cells per animal were counted, the heterophil/lymphocyte ratio (H/L) was calculated by dividing the number of heterophils by the number of lymphocytes [22].

### 2.6. Cecal Microbiota

To measure the cecal bacteria counts, the cecum collected at the end of the experiment was immediately placed in an anaerobic jar (Oxoid Limited, Basingstoke, Hampshire, UK) equipped with an aerogen sachet (Oxoid Limited, Basingstoke, Hampshire, UK). The cecum (1.0 g) was homogenized with 10 mL of sterilized 0.1 M phosphate-buffered saline at pH 7.0. The homogenate was serially diluted from an initial 10^1^ to 10^−9^. Then, 100 µL of each diluted sample was plated on Difco™ *Lactobacilli* MRS agar for *Lactobacillus*, nutrient agar for total aerobic bacteria, and Violet red bile agar with MUG (Alpha Biosciences Inc., Baltimore, MD, USA) for *E. coli* and coliforms, respectively. The *Lactobacillus* was incubated for 48 h at 37 °C under anaerobic conditions. Coliforms, *E. coli*, and the total aerobic bacteria were incubated for 24 h at 37 °C under aerobic conditions. The data are presented as log 10 colony-forming units (CFU) per gram of cecal content.

### 2.7. Animal Behavior and Welfare Indicators

Observations of the behaviors and welfare of broiler chickens were based on the scientific concepts of animal behavior and welfare as per the Welfare Quality^®^ Consortium [23]. Animal behaviors, such as drinking, eating, locomotion (movement), grooming, and resting (lying down), were observed on Days 25 (06:00–16:00), 30 (10:00–18:00), and 33 (12:00–20:00), for each replicate group via ceiling-mounted infrared video. The data obtained by three observations are expressed as the average value. These observations entailed determining the actions performed by each bird in the camera’s field of view at each time interval. The images of the recorded video were scanned and measured at 20 min intervals to analyze the behavioral indicators of the animals. From these observations, behavioral parameters [17] for drinking, eating, locomotion (movement), grooming (preening), and resting (lying down, sitting position) were calculated. A total of three people, including the author, an animal caretaker, and a researcher, participated in assessing the procedures, and the average values were measured based on data obtained from images of the recorded videos (see above). The assays of behavioral patterns for reliability analysis in animal behavior research include inter- and intra-assay variabilities of 5.0%. The foot pads and hock burns scores were assessed on a 5-point scale (0, no lesion; 1, superficial lesion; 2, superficial lesion > 0.5 cm; 3, deep lesion > 1.0 cm; 4, whole hock extensively altered) [24]. The plumage (feather scoring) on the back and chest was classified on a 4-point scale (0, clear and fluffy; 1, slightly dirty < 15–25%; 2, moderately dirty 25–50%; 3, completely dirty > 50%) [24]. The gait score was assessed on a 6-point scale (0, fluent walking without detectable abnormality for chickens; 1, slight undefined defect in gait; 2, definite changes/defects and waddling walk; 3, clearly fluent gait restriction; 4, severe gait defects or difficulty moving; 5, unable to walk [25]. The foot pads, hock burns, plumage, and gait scores were determined by directly checking with the naked eye along with camera observation on the same day (see above).

### 2.8. Statistical Analysis

Statistical analysis was performed by using a PROC GLM procedure in SAS Version 9.2. [26] under a completely randomized design in 2 × 2 factorial arrangements. A 2 × 2 factorial design was used to analyze the data of performance as a response to 2 levels of the poultry system (i.e., conventional, digital poultry system) supplemented with prebiotics (0, 500 ppm of diet). All data were tested for distribution normality and the homogeneity of variance. The experimental unit was the pen. The effects of the probiotics and digital poultry system were set as the main effects and interactions. The differences in the mean values between the treatment groups were analyzed using two-way analysis of variance (ANOVA). It was confirmed that the trend observed in the dependent variable was normally distributed in the population, and it was assumed that the obtained sample was also normally distributed. Thus, the conclusions derived from the experiments are reliable. The significant differences among treatment means were determined using Tukey’s test (*p* < 0.05).

The model used is:Xij = μ + αi + βj + αβij + εijk
where Xij = an observation, μ = the overall mean, αi = main effect of probiotics, βj = main effect of digital poultry system, αβij = interaction effect between probiotics and digital poultry system, and εijk = random error associated with each observation.

## 3. Results

### 3.1. Growth Performance and Organ Index

Significant differences in the growth performance were observed during the entire experiment because of the main effects of a probiotics (*p* < 0.013 to *p* < 0.025) or a digital poultry system (*p* < 0.012 to *p* < 0.015). There was an interaction effect between the probiotics and the digital poultry system (*p* < 0.015 to *p* < 0.020) (Table 2). The body weight, feed intake, and feed conversion ratio of broiler chickens were significantly improved in animals reared under the combination of the probiotics and the digital poultry system compared to the conventional poultry system without the probiotics. The results showed no difference between the DPCS and CON500 groups. No animal mortality was observed during the entire experimental period.

The immune organ indexes, such as the spleen, thymus, and bursa of Fabricius, were significantly higher in the group of the combination of the probiotics and the digital poultry system than those in the other groups, as well as higher in the DPCS and CON500 groups than the CON group because of the main effects of the probiotics (*p* < 0.011 to *p* < 0.027) and the digital poultry system (*p* < 0.007 to *p* < 0.020) or the interaction effect between the probiotics and the digital poultry system (*p* < 0.020 to *p* < 0.031); however, there were no differences between the DPCS and CON500 groups (Table 3).

### 3.2. Immune Biomarkers

The immune biomarkers of the broiler chickens were significantly the highest in the DLS500 group raised with the combination of the probiotics and the digital poultry system because of the main effects of the probiotics (*p* < 0.008 to *p* < 0.031) or the digital poultry system ((*p* < 0.012 to *p* < 0.020). The interaction between the probiotics and the digital poultry system also had a significant effect on the immune function (*p* < 0.011 to *p* < 0.031) (Table 4). Furthermore, the DPCS and CON500 groups showed significant improvement compared to the conventional poultry system without the probiotics (*p* < 0.05). In contrast, there was no difference between the DPCS and CON500 groups. The serum IgG levels were significantly higher in the DPS500 group than the other groups (*p* < 0.05), and there was no difference between the DPCS and CON500 groups. The serum corticosterone levels and heterophil to lymphocyte ratios (H/L) were significantly highest in the CON and lowest in the DPS500 groups (*p* < 0.05), and no differences were noted between the DPS500 and DPCS groups.

### 3.3. Cecal Microbiota

The cecal microbiota had significant differences observed at the end of the experiment (35 day) because of the main effects of the probiotics (*p* < 0.019 to *p* < 0.031) or the digital poultry system (*p* < 0.010 to *p* < 0.025). There was an interaction effect between the probiotics and digital poultry system (*p* < 0.015 to *p* < 0.031) (Table 5). Cecal *Lactobacillus* was significantly higher with the combination of the probiotics and the digital poultry system than the conventional livestock system; contrarily, the coliforms, *E. coli*, and total aerobic bacteria were low (*p* < 0.05), while the cecal microbiota showed no differences between the CON500 and DPCS groups.

### 3.4. Animal Behavior and Welfare

The behavioral parameters, such as eating, locomotion (moving, walking), drinking, resting (lying down), and grooming, as well as the body condition scores of the foot pads, hock burns, plumage, and gait of the broiler chickens were significantly higher in the order of the DPS500, DPCS, CON500, and CON groups because of the main effects of the probiotics (*p* < 0.015 to *p* < 0.025) or the digital poultry system (*p* < 0.010 to *p* < 0.030). In addition to, there was an interaction effect between the probiotics and digital poultry system (*p* < 0.015 to *p* < 0.030) (Table 6 and Table 7). However, there were no differences between the DPCS and CON500 groups.

## 4. Discussion

Renewed interest in the application effects of a digital poultry system along with probiotics has gained global attention in recent years owing to the convergence of livestock and ICT tools as sustainable livestock farming strategies in response to climate change [1,2,9,11]. The main goal of this study was to identify the biological mechanisms of the growth performances, behaviors, and welfare of broiler chickens reared by the combination of a probiotic and a digital poultry system. The research findings highlight a new fact that the growth performances of broiler chickens could be improved by the combination of a probiotics and a digital poultry system compared to the conventional poultry system (Table 2). The results of this study are partially in line with the authors’ previous report, wherein improved egg production was observed for laying hens that were not fed probiotics and housed in a digital poultry system converged with ICT [9,14], along with improved growth performances of broiler chickens housed in a digital poultry system without being fed probiotics [27], and enhanced swine production when raised under a combination of a probiotics and a digital livestock system compared to the conventional system [6,9,10]. These observations support our results, but studies on the growth performances of broiler chickens reared under the combined system have not been reported yet. Since the combination of a probiotics and a digital poultry system helped improve the growth performance of broiler chickens, it can be considered as a molecular mechanism related to animal biomarkers through balanced immune organ indexes, such as the spleen, thymus, and bursa of Fabricius, as well as the serum IgG, corticosterone, H/L ratio, and cecal bacteria [3,9,14,27] (Table 3, Table 4 and Table 5). The fact that the corticosterone level was highest in the CON group may be attributable to physiological changes in the broiler chickens due to insufficient basic animal management techniques, such as environmental temperature, humidity, ventilation, and maintenance of dietary intake and water supply [4,5,9,14,15]. In particular, compared to the digital poultry system groups, the animal caretaker’s frequent entry into the livestock barn and interference with animal management to observe the condition of the livestock, such as animal feeding management and animal care, would have acted as major stress factors [8,9,12,15,16,17]. In the CON group, when the water supply stopped suddenly, the ventilation fans stopped working so that the environmental abnormalities caused abnormal animal movements, which we observed directly and took action on. Meanwhile, the digital poultry system groups used automation technology combining big data and ICT tools to identify the housing environment and animal status without time and space constraints, allowing real-time monitoring to quickly implement appropriate measures so as to ensure comfortable animal welfare. With these improvements, it can be seen that the corticosterone levels were lower compared to those of the CON group [8,15,16,17]. The digital poultry system group used automation technology combining big data and ICT tools to identify the housing environment and animal status without time and space constraints, allowing real-time monitoring via remote control using mobile phones to quickly implement appropriate measures so as to ensure comfortable animal welfare. With these improvements, it can be seen that it affects the improvement of biomarkers, the development of immune organs, the level of antibodies and corticosterone, etc. [8,15,16,17]. The digital poultry system group used automatic technology to monitor the housing environment, feeding management, and animal movement in real-time, such that appropriate actions can be taken when problems arise (see above). Therefore, by increasing the dietary intake and nutrient digestibility (not determined here [8]), the development of immune cells is stimulated while the serum IgG, corticosterone, H/L ratio, and cecal bacteria become balanced as the biomarkers are maintained (Table 3, Table 4 and Table 5). By maintaining the balance of biomarkers and promoting animal welfare, it is expected that the growth performance could be improved [1,6,8,9,10,11,12]. On the other hand, probiotics that can replace antibiotics in poultry feed are known to compete with other gut bacteria for nutrients, induce the production of antibacterial substances, and improve the growth performances of birds by increasing their immune functions, nutrient digestibility, intestinal morphologies, and microbiota [3,14,27,28]. The significantly higher growth performance of broiler chickens in the DPS500 group could be due to the interaction of the combination of a probiotics and a digital poultry system. All these are related to the effects of probiotics.

The study results indicate that the combined system can suppress animal stressors by stimulating immune functions in broiler chickens. In addition, one of the most-important roles of probiotics is to stimulate immunity against invading pathogenic bacteria; they are known to improve the appropriate microbial environment of the digestive tract and host animal immunity [3,29,30]. Animals raised under good animal feeding and environmental management practices using ICT show lower stress levels, which ultimately improves animal production [1,16,31]. The fact that the combination of a probiotics and a digital poultry system improves animal biomarkers, such as the development of immune organs, the levels of serum IgG and corticosterone, the H/L ratio, and the control of cecal bacteria, is consistent with the results of previous reports and digital poultry systems fused with ICT [14,27]. It is clear how the probiotics exert their effect. The digital poultry system uses automatic technology to monitor the housing environment, feeding management, and animal movement in real-time, such that appropriate actions can be taken when problems arise (see above). Therefore, by increasing the dietary intake and nutrient digestibility (not determined here [8]), the development of immune cells is stimulated while the serum IgG, corticosterone, H/L ratio, and cecal bacteria balance as biomarkers are maintained (Table 3, Table 4 and Table 5). By maintaining the balance of biomarkers and promoting animal welfare, it is expected that the growth performance could be improved [1,6,8,9,10,11,12].

These results found with respect to broiler chickens may be attributable to the use of smartphones through sensing platforms, automatic animal feeding management, and environmental management via central control through cloud computing systems [1,7,10]. It can be observed that probiotics stimulate the development and growth of immune cells in conventional poultry systems based on the fact that the CON500 group had better immune function than the CON group, even though there was no difference between the DPCS and CON500 groups. Probiotics are well known to enhance the development and growth of immune cells by feed intake stimulation in the digestive tract (Table 2), as well as increase the absorption and utilization rates of nutrients (not determined) [3,6].

Our results suggest that the intestinal microbiota balance was well maintained in broilers raised under a combination of a probiotics and a digital poultry system. The gut microbiota are directly related to the growth performance and health of animals. In general, the number and structural composition of gut microbiota play important roles in the absorption of nutrients and health in the host animals [32]. There were no differences between the DPCS and CON500 groups in terms of the cecal *Lactobacilli*, *E. coli*, and total aerobic bacterial balance, but the DPCS and CON500 group showed more balanced results than the CON group. The digital poultry system maintains gut microbiota balance compared to the CON group (Table 5). The gut microbiota are closely related to the health of the host and are greatly influenced by diet, animal management, and environment [2,3,4,6,27,28,29]. Animals raised in the digital poultry system may have achieved maintenance of gut microbiota via a reduced number of harmful bacteria and an increased number of beneficial bacteria in the cecum because of their increased nutrient digestibility and good environmental management through digital technologies (remote control and sensing technologies), which can help control these factors well. The balanced gut microbiota of livestock can have positive effects on the survival and growth of these animals, such as improved immune functions and body weight gain [1,8,28,29]. These research findings could be a beneficial effect of the combination of a probiotics and a digital poultry system and may be due to the biological function of the probiotics and the automated animal feeding and environmental management from the digital poultry system. The benefits are believed to be derived from the stimulated immune functions (Table 3 and Table 4) and activated nutrient metabolisms (not determined) in the digestive tracts of animals housed under a combination of a probiotics and a digital poultry system [6]. The results are also in line with the authors’ previous reports [6,14,27]. The use of either probiotics or raising animals alone under a digital poultry system without the use of a probiotics is known to maintain balanced cecal microflora, but few results have been reported in the literature for broiler chickens so far [6]. Furthermore, the result that the cecal microbiota balance of broiler chickens reared in the ICT convergence smart system was maintained supports this result [27]. Probiotics are a class of beneficial microbiota that influence host responses by regulating appropriate gut microbiota. In broiler chickens, the cecal environment can be influenced by the ingestion of probiotics that prevent gut colonization by harmful bacteria [3,30,31]. The growth performances of broilers reared under the conventional poultry systems were lower than those of broilers reared under a combination of probiotics and digital poultry systems, and the CON group showed lower values than CON500 (Table 2). Even though the control group in the experiment was not exposed to specific stress factors, the fact that the microbial balance and immune functions in the cecum were worse than those of the other treatment groups can be seen as a synergistic effect of the combination of probiotics and the digital poultry system (Table 3, Table 4 and Table 5). *Lactobacillus* sp. is known to maintain antibacterial activity and bacterial balance, in addition to maintaining the natural stability of the intestinal microflora against biological changes caused by various environmental conditions [2,3]. In the conventional poultry system, poor animal health, such as decreased nutrient digestibility, as well as lower immune function due to insufficient management of the housing environment and animal feeding by the animal caretaker, may have disturbed the stability of the intestinal microbial ecosystem and perturbed homeostasis, causing the decrease in growth performance [2,3,28,30]. The digital poultry system utilizes mobile phones and sensing technology to proactively monitor animal movements and environmental conditions to solve problems related to animal production in real-time, thereby increasing nutrient digestibility and improving the immune system development and biomarker balance in the animals. This may have contributed to increasing the growth performance by maintaining homeostasis [1,6,9,10,11]. The results of the present study show that probiotics improve immune functions while relieving stress by maintaining microflora balance and inhibiting the growth of harmful bacteria in the cecum. In particular, *Lactobacillus* is known to maintain antibacterial activity and bacterial balance, in addition to maintaining the natural stability of gut microbiota against biological changes caused by various types of animal stress. Specific measures were established to frequently observe the farm environment and chicken movements directly in the CON groups, monitor the digital poultry system groups via remote control in real-time to solve problems, and collect information on the behavioral patterns and health conditions of the animals. During the experiments, if the environmental conditions changed or problems were noted in terms of floor moisture and ammonia because of the water supply or bedding, the animal caretaker directly entered the cage and operated the supplied equipment or replaced the bedding to solve the problem. When lesions or inflammation of the foot pads were identified, the wounds were treated with disinfectant to help recovery (see above). In stressful environments such as poor health of the animals, heat and cold, digestive disorders, reduced nutrient availability, and reduced immune function [3,14,15], stabilization of the intestinal microbial ecosystem is disturbed and homeostasis is broken, resulting in reduced growth performances of the animals [14,29,32]. It is probably for this reason that the growth performances of broiler chickens housed under the conventional poultry system were lower than those housed under the combination of a probiotics and a digital poultry system (Table 2), with the CON group showing lower values than the CON500 group. The digital poultry system, a convergence technology of ICT and livestock farming, is known to lower animal stress. In particular, it reduces animal stress by appropriately maintaining the factors related to growth performance, increases nutrient digestibility, and enhances the development of immune organs via preemptively monitoring animal movements and environmental conditions using mobile phones and sensing technology [6,33].

The study results showed that the combination of a probiotics and a digital poultry system can improve animal behavioral parameters, such as drinking, eating, locomotion (moving), grooming, and resting (lying down) as well as animal welfare indexes, such as the foot pads, knee burns, plumage, and gait, compared to the conventional poultry system (Table 6 and Table 7). Despite maintaining appropriate animal stocking density during the experiment, there were significant differences in the foot pads, knee burns, plumage, and gait between the CON and the other groups. The reason for this is that the CON group had an inadequate environment, feeding management, and animal management; on the other hand, the digital poultry system groups allowed the collection and utilization of standardized data using a digital system combining big data and ICT, through which we were able to identify abnormal entities in real-time via remote control and implement specific measures. Broiler chickens are often raised on flat floors, and bedding containing sawdust is provided to create an environment where the livestock can feel comfortable. In the process of raising livestock, the amount of manure discharge increases, and the moisture content of the floor increases gradually. Physical abnormalities, such as dermatitis on the soles of the foot pads and feather pulling, can easily occur in animals that come into direct contact with wet floors. This may have affected the foot pads, knee burns, plumage, and gait scores in the CON group. The digital poultry system operates in a self-driving manner using an automatic litter spreader according to input information previously set by the manager. By monitoring the condition of the bedding and animal in real-time, specific measures can be implemented, which can improve animal welfare and comfort. As the digital poultry system groups had better environment and animal management than the CON groups, there were fewer lesions on the soles of the foot pads of the broiler chickens, and the other parameters (knee burn, plumage, gait) were also observed to be superior. The authors’ previous study on an ICT-converged digital poultry system improving egg production by maintaining good animal behavioral welfare in laying hens also supports these findings [9,14]. In their previous studies, the authors reported that animal behavioral welfare indicators, such as drinking, eating, locomotion (moving), grooming, and resting (lying down), appearance, feather conditions, body conditions, and the health of birds reared in an ICT-converged digital poultry system in the absence of probiotics were improved compared to those under the conventional livestock system [6,9,34]; however, the results of research on animal behavioral welfare under the combination of a probiotics and a digital poultry system with converged ICT in broiler chickens are not well known. The improved animal behavioral welfare observed in this study may be attributed to the superior retention of biomarkers that maintain animal activity, including immune function and cecal microflora, through controlled animal feeding management and environmental stress factors via a remote system (Table 4 and Table 5) [6,9,33,34]. Factory-intensive conventional livestock systems increase animal stress due to various reasons including poor environmental conditions (hot and cold), health deterioration due to careless animal management, digestive disorders, low nutrient availability, and a decreased immune function [6,35], which threatens animal behavioral welfare and carbon neutrality. Factory-intensive livestock systems threaten carbon neutrality by increasing odor and methane generation due to poor feeding environments and livestock management systems. The digital livestock system including poultry, which uses ICT convergence with farm animals, can realize carbon neutrality in the livestock industry by establishing a low-carbon livestock management system. The digital livestock system can help realize carbon neutrality through the development of methane-reducing feed with the simultaneous establishment of a low-carbon livestock management system [6,9,11].

An ICT-converged digital poultry system using AI, big data, cloud computing, IOT, ML, and multimodal sensor technologies can guarantee greater amounts of free animal rights by reducing stress, which is related to animal behavioral welfare [1,6,7,9,10,11]. It could reduce animal stress by using remote control and real-time monitoring of the behavioral welfare, care, and movements of broiler chickens. They can observe animal feeding, the barn environment (temperature, humidity, ventilation), and the health management of individuals by remote control and sensing technology through the use of the Internet and mobile phones [14,17,27]. To meet the consumption demands of animal-based foods to solve the growing food procurement problems of humans, existing breeding programs and animal feeding practices focused on high livestock production and economic productivity have influenced not only the health and behavioral welfare of broiler chickens, but also other farm animals. These can lead to several stressors that affect animal wellbeing [15,16,17]. Probiotics can inhibit colonization by harmful bacteria and help activate beneficial bacteria in the gut. In particular, balanced cecal microbiota in the digestive tracts of poultry through the use of probiotics can help improve animal behavioral welfare by maintaining microbial homeostasis [3,6,36,37]; as seen from Table 6, the fact that the CON500, DPCS, and DPS500 groups had better animal behavioral welfare scores and body condition scores than the CON group supports this reason. Intensive animal feeding management practices for poultry may cause stress and adversely affect animal behavioral welfare, which are important problems related to economic losses in broiler farms, but the use of probiotics in poultry can improve livestock production by maintaining good animal behavioral welfare [3,38,39,40]. The main result of this study was that, in the conventional poultry system of raising chickens, the farmers must adhere to good production practices, which include providing the chickens with good ambient conditions, adequate stocking density (greater than 10 chickens per square meter), the availability of food and water, and sufficient living space, to meet the welfare conditions. The difference here compared to the digital poultry system is that, with the conventional poultry system, a person takes care of everything, while in the digital system, all work is handled almost automatically. Therefore, it can be considered that the control group was exposed to numerous stress-causing factors (environment, animal feeding management, animal movement, etc.), so that the production and immunological parameters were significantly worse than those of chickens raised under the digital poultry system.

## 5. Conclusions

In conclusion, the study found that the combination of a probiotics and a digital poultry system could improve animal behavioral welfare and the growth performance of broiler chickens, as reported in the text. The results showed that the combined system was able to maintain an animal homeostasis balance such as immune biomarkers and cecal microbiota. In addition, growth performance was improved by enhancing the behavioral welfare scores and body condition scores in the broiler chickens. Although the study did not find a significant effect between the conventional poultry system with the probiotics and the control group of the digital poultry system without probiotics, there was a better balance of animal homeostasis in broiler chickens raised using the probiotics in the digital poultry system. This can be seen as a biological mechanism that appeared through the balanced interaction between the functional characteristics of the probiotics and the enabling of livestock management and environmental control monitoring through remote control and sensing technology by the digital poultry system, which is a convergence technology of ICT and livestock farming.

## Figures and Tables

**Figure 1 microorganisms-11-02345-f001:**
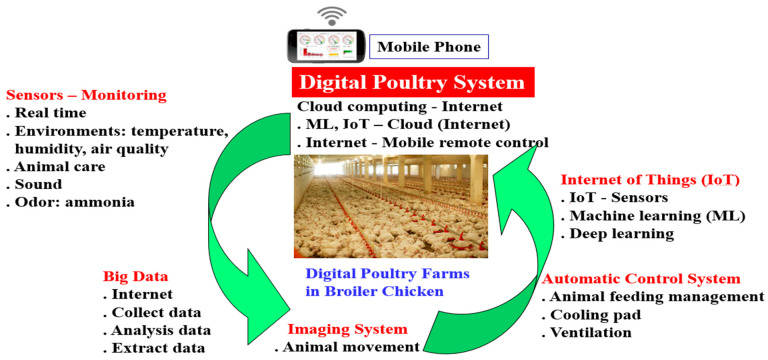
Flowchart of the digital poultry system with the convergence of ICT and poultry farming.

**Table 1 microorganisms-11-02345-t001:** Formula and chemical composition of the probiotic diet (% as-fed basis).

Ingredients	Grower (Days 1–21)	Finisher (Days 22–35)
Yellow corn	52.00	50.00
Soybean meal	34.00	25.00
Corn gluten meal	4.70	5.70
Wheat bran	0	10.00
Beef tallow	5.00	5.00
Limestone	1.25	1.25
Dicalcium phosphate	1.70	1.70
Salt	0.25	0.25
DL-methionine	0.30	0.30
L-lysine-HCL	0.30	0.30
Mineral premix ^a^	0.34	0.34
Vitamin premix ^b^	0.16	0.16
Probiotic mixture ^c^ Chemical composition	0.05	0.05
Metabolizable energy, Kcal/kg	3100	3150
Crude protein	22.0	20.0
Lysine	1.32	1.15
Methionine	0.52	0.50
Methionine + cysteine	0.78	0.73
Calcium	1.00	1.00
Available phosphorous	0.45	0.40

^a^ Supplied per kilogram of diet: Fe (ferrous sulfate), 80 mg; zinc (zinc oxide), 80 mg; Mn (manganese sulfate), 70 mg; Cu (copper sulfate), 7 mg; I (calcium iodate), 1.20 mg; Se (sodium selenite), 0.30 mg; Co (cobalt), 0.70 mg. ^b^ Supplied per kilogram of diet: vitamin A (retinyl acetate), 10,500 IU; vitamin D_3_ (cholecalciferol), 4100 IU; vitamin E (dl-α-tocopheryl acetate), 45 mg; menadione, 3.0 mg; thiamin, 2.5 mg; riboflavin, 5.0 mg; pyridoxine, 4.0 mg; cyanocobalamin, 0.02 mg; niacin, 44 mg; pantothenic acid, 17 mg; folic acid, 1.5 mg; biotin, 0.18 mg. ^c^ The probiotic mixture contained 3.5 × 10^8^ CFU/g of *Bacillus subtillus*, *Streptomyces galilaeus*, and *Sphingobacteriaceae*.

**Table 2 microorganisms-11-02345-t002:** Growth performance by using the combination of the probiotics and the digital poultry system in broiler chickens during the entire experiment ^1^.

Items	Conventional System		Digital Poultry System		SEM ^2^	*p*-Value ^3^		
	CON	CON500	DPCS	DPS500		P	D	P × D
Grower, (d 1–21)								
Body weight, g/bird	841 ^c^	860 ^b^	858 ^b^	876 ^a^	11.58	0.020	0.031	0.017
Feed intake, g.bird	980 ^c^	997 ^b^	1002 ^b^	1022 ^a^	12.01	0.019	0.020	0.030
Feed conversion ratio	1.16	1.16	1.17	1.17	0.010	0.072	0.103	0.097
Finisher (d 22–35)								
Body weight, g/bird	1085 ^c^	1213 ^b^	1199 ^b^	1310 ^a^	18.78	0.030	0.022	0.018
Feed intake, g.bird	2015 ^c^	2116 ^b^	2100 ^b^	2191 ^a^	27.40	0.022	0.016	0.009
Feed conversion ratio	1.85 ^a^	1.74 ^b^	1.75 ^b^	1.67 ^c^	0.010	0.010	0.015	0.019
Overall (d 1–35)								
Body weight, g/bird	1926 ^c^	2073 ^b^	2057 ^b^	2186 ^a^	26.76	0.013	0.025	0.016
Feed intake, g.bird	2995 ^c^	3113 ^b^	3102 ^b^	3213 ^a^	41.26	0.025	0.012	0.020
Feed conversion ratio	1.56 ^a^	1.50 ^b^	1.51 ^b^	1.47 ^c^	0.008	0.017	0.020	0.015

^1^ CON: control group with a conventional poultry system without probiotics, CON500: conventional poultry system with 500 ppm of probiotics, DPCS: digital poultry control system without probiotics, and DPS500: digital poultry system with 500 ppm of probiotics. ^2^ SEM: standard error of means. ^3^ P: probiotics, D: digital poultry system, P × D: interaction of probiotics and digital poultry system. ^a,b,c^ Means within the same row with different superscripts differed significantly (*p* < 0.05).

**Table 3 microorganisms-11-02345-t003:** Immune organ index by using the combination of the probiotics and the digital poultry system in broiler chickens at the end of the experiment (35 days) (g/100 g live body weight) ^1^.

Items	Conventional System		Digital Poultry System		SEM ^2^	*p*-Value ^3^		
	CON	CON500	DPCS	DPS500		P	D	P × D
Spleen	0.10 ^c^	0.13 ^b^	0.13 ^b^	0.15 ^a^	0.002	0.011	0.020	0.026
Thymus	0.11 ^c^	0.16 ^b^	0.16 ^b^	0.20 ^a^	0.002	0.015	0.007	0.020
Bursa of Fabricius	0.08 ^c^	0.11 ^b^	0.12 ^b^	0.16 ^a^	0.001	0.027	0.017	0.031

^1^ CON: control group with a conventional poultry system without probiotics, CON500: conventional poultry system with 500 ppm of probiotics, DPCS: digital poultry control system without probiotics, and DPS500: digital poultry system with 500 ppm of probiotics. ^2^ SEM: standard error of means. ^3^ P: probiotics, D: digital poultry system, P × D: interaction of probiotics and digital poultry system. ^a,b,c^ Means within the same row with different superscripts differed significantly (*p* < 0.05).

**Table 4 microorganisms-11-02345-t004:** Immune biomarkers by using the combination of the probiotics and the digital poultry system in broiler chickens at the end of the experiment (35 days) ^1^.

Items	Conventional System		Digital Poultry System		SEM ^2^	*p*-Value ^3^		
	CON	CON500	DPCS	DPS500		P	D	P × D
Corticosterone, ng/mL	2.17 ^a^	1.24 ^b^	1.03 ^b^	0.50 ^c^	0.017	0.031	0.020	0.011
IgG, μg/dL	56.59 ^c^	67.72 ^b^	68.63 ^b^	88.06 ^a^	0.752	0.022	0.017	0.025
Heterophil (H), %	19.28 ^a^	17.21 ^b^	16.88 ^b^	15.64 ^c^	0.231	0.027	0.012	0.018
Lymphocyte (L), %	66.72 ^c^	70.66 ^b^	70.80 ^b^	75.17 ^a^	0.920	0.008	0.015	0.031
H/L ratios	0.29 ^a^	0.24 ^b^	0.23 ^b^	0.21 ^c^	0.001	0.012	0.015	0.022

^1^ CON: control group with a conventional poultry system without probiotics, CON500: conventional poultry system with 500 ppm of probiotics, DPCS: digital poultry control system without probiotics, and DPS500: digital poultry system with 500 ppm of probiotics. ^2^ SEM: standard error of means. ^3^ P: probiotics, D: digital poultry system, P × D: interaction of probiotics and digital poultry system. ^a,b,c^ Means within the same row with different superscripts differed significantly (*p* < 0.05).

**Table 5 microorganisms-11-02345-t005:** Cecal microbiota by using the combination of the probiotics and digital poultry system in broiler chickens at the end of the experiment (35 days) (log_10_ CFU/g) ^1^.

Items	Conventional System		Digital Poultry System		SEM ^2^	*p*-Value ^3^		
	CON	CON500	DPCS	DPS500		P	D	P × D
*Lactobacillus*	6.47 ^c^	7.05 ^b^	7.33 ^b^	7.70 ^a^	0.066	0.009	0.010	0.008
*E. coli*	6.75 ^a^	6.03 ^b^	5.83 ^b^	5.07 ^c^	0.074	0.011	0.005	0.010
Coliform bacteria	7.60 ^a^	6.55 ^b^	6.42 ^b^	5.88 ^c^	0.060	0.025	0.010	0.005
Total aerobic bacteria	7.85 ^a^	7.07 ^b^	6.89 ^b^	6.03 ^c^	0.090	0.001	0.005	0.001

^1^ CON: control group with a conventional poultry system without probiotics, CON500: conventional poultry system with 500 ppm of probiotics, DPCS: digital poultry control system without probiotics, and DPS500: digital poultry system with 500 ppm of probiotics. ^2^ SEM: standard error of means. ^3^ P: probiotics, D: digital poultry system, P × D: interaction of probiotics and digital poultry system. ^a,b,c^ Means within the same row with different superscripts differed significantly (*p* < 0.05).

**Table 6 microorganisms-11-02345-t006:** Behavioral frequency by using the combination of the probiotics and the digital poultry system in broiler chickens ^1^.

Items	Conventional System		Digital Poultry System		SEM ^2^	*p*-Value ^3^		
	CON	CON500	DPCS	DPS500		P	D	P × D
Drinking	18.85 ^a^	17.22 ^b^	17.38 ^b^	14.72 ^c^	0.0221	0.023	0.030	0.022
Eating	33.86 ^c^	36.83 ^b^	37.01 ^b^	38.87 ^a^	0.364	0.015	0.025	0.015
Locomotion	19.68 ^c^	22.05 ^b^	21.90 ^b^	23.16 ^a^	0.282	0.025	0.021	0.030
Grooming	10.76 ^c^	11.57 ^b^	11.89 ^b^	13.22 ^a^	0.137	0.018	0.010	0.018
Resting	16.85 ^a^	12.33 ^b^	12.82 ^b^	10.03 ^c^	0.169	0.015	0.023	0.023

^1^ CON: control group with a conventional poultry system without probiotics, CON500: conventional poultry system with 500 ppm of probiotics, DPCS: digital poultry control system without probiotics, and DPS500: digital poultry system with 500 ppm of probiotics. ^2^ SEM: standard error of means. ^3^ P: probiotics, D: digital poultry system, P × D: interaction of probiotics and digital poultry system. ^a,b,c^ Means within the same row with different superscripts differed significantly (*p* < 0.05).

**Table 7 microorganisms-11-02345-t007:** Body condition scores by using the combination of the probiotics and the digital poultry system in broiler chickens ^1^.

Items	Conventional System		Digital Poultry System		SEM ^2^	*p*-Value ^3^		
	CON	CON500	DPCS	DPS500		P	D	P × D
Foot pads	2.61 ^a^	2.02 ^b^	2.07 ^b^	1.68 ^c^	0.048	0.018	0.010	0.025
Knee burns	2.57 ^a^	2.04 ^b^	1.83 ^b^	1.41 ^c^	0.055	0.017	0.022	0.019
Plumage	2.84 ^a^	2.40 ^b^	2.38 ^b^	2.01 ^c^	0.032	0.020	0.020	0.025
Gait	2.33 ^a^	1.82 ^b^	1.77 ^b^	1.39 ^c^	0.050	0.011	0.019	0.028

^1^ CON: control group with a conventional poultry system without probiotics, CON500: conventional poultry system with 500 ppm of probiotics, DPCS: digital poultry control system without probiotics, and DPS500: digital poultry system with 500 ppm of probiotics. ^2^ SEM: standard error of means. ^3^ P: probiotics, D: digital poultry system, P × D: interaction of probiotics and digital poultry system. ^a,b,c^ Means within the same row with different superscripts differed significantly (*p* < 0.05).

## Data Availability

The data and analyses presented in this paper are freely available from the corresponding author.

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
