# Peer review of "Impact of the Combination of Probiotics and Digital Poultry System on Behavior, Welfare Parameters, and Growth Performance in Broiler Chicken"

_microorganisms, 2023, doi:10.3390/microorganisms11092345_

Round 1
Reviewer 1 Report
The authors present data from a study titled „Impact of the Combination of Probiotics and Digital Poultry System on Behavior, Welfare Parameters, and Growth Performance in Broiler Chicken“. The results show that the combined system (digital poultry system and probiotics) can improve broiler growth performance and animal behavioral welfare.
The introduction is longer, but it presents all the important information about the topic. The methodology is appropriate for the set goals. Statistical analysis is appropriate, study design and execution are scientifically sound. The interpretation of the results, their commenting, and discussion are not sufficiently grounded in concrete facts and mechanisms of action.
Specific comments and suggestions:
L 310 – L 312 How do authors explain the corticosterone level in the control group? Why was there such a difference? What specific stressors during this experiment caused corticosterone to be significantly higher than the other groups? This finding needs to be described in more detail in the discussion.
L 352 - L 354 How is it possible that at a stocking density of 10 birds/m2, there was such a large difference between the control group and the DSC group? In what way did the digital system influence the chickens to have fewer lesions on foot pads? Also with other parameters (knee burns, plumage, gait). State-specific facts and describe them in the discussion.
L 385 – L 387 It is clear how the probiotics exert their effect, but how do you explain the effect of the digital system with concrete facts from your experiment?
L 388 – L 389 All this relates to the effect of probiotics.
L 394 – L 398 It is still not clearly stated how the digital system affects the improvement of biomarkers, the development of immune organs, the level of antibodies and corticosterone, etc.
L 408 – L 415 This positive effect can be attributed to the probiotics. How did the digital system contribute to a better gut microbiota in your experiment?
L 434 – L 441 The control group in your experiment was not exposed to any specific stressors. In your experiment, you did not simulate some of the stressors that might occur in conventional farms. This paragraph cannot be linked to your results.
L 441 – L 446 This explanation is not based on specific factors if you only monitored the chickens and did not take any specific measures. If you took any specific measures during the experiment with the chickens in the digital system, you must describe those measures. For example, you followed the movement of animals and got some data. Did you then take any measures with the chickens in the digital system to stimulate them to move more or less? If so, please list them. For example, during the experiment, you found that there is some problem with the ventilation or temperature of the chickens in the digital system. What measures were taken then? If so, aren't such corrective measures also taken with chickens in a conventional housing system? What is the difference?
In the conventional system of keeping chickens, farmers must adhere to good production practices, which include providing the chickens with good ambient conditions, adequate stocking density (which is greater than 10 chickens per square meter), availability of food and water, and sufficient living space to meet welfare conditions. The difference compared to the digital system is that with the conventional system, a person takes care of everything, while with the digital system, it should work almost automatically. Now the question arises whether in your experiment the control group was exposed to so many stressogenic factors (you should specify which ones), that the production parameters and immunological parameters were significantly worse compared to the chickens in the digital system.
If in your experiment there were stressogenic factors in the control group, they should be mentioned, to better understand and analyze the results, that is, to better understand the effects of the digital system.

Reviewer 2 Report
In general, the study entitled “ Impact of the Combination of Probiotics and Digital Poultry System on Behavior, Welfare Parameters, and Growth Performance in Broiler Chicken” is really interesting, well described and discussed. My minor comments aim to increase the scientific soundness and clarity of it.
My comments:
Introduction – is too long. Please delete superfluous parts.
Line 96 – please explain what is “physiological welfare”. Could “animals’ welfare” be even physiological or non-physiological?
Line 192 – please specify which immune organs were dissected out.
Line 217 – please explain what MUG stands for.
Line 236 – how many researchers participated in assessing procedures? Did they average obtained results somehow?
Line 254 - did the authors checked the normality assumption for ANOVA?
Line 285, 293 etc. – please remove double coma.
Line 531 – Reference list is in total mess. Please correct it to meet the Journal criteria. In the body text the last reference is numbered as [48] while in reference list the last one has number [55].
Manuscript need moderate English language revision.
Round 2
Reviewer 1 Report
The authors answered all questions in detail. They described and clarified all doubts in the manuscript and now it is completed and in this form acceptable for publication.
I thank the authors for their extensive responses.
I suggest that the manuscript be accepted for publication.